# Red Cell Distribution Width as a Prognostic Indicator in Acute Medical Admissions

**DOI:** 10.3390/jcm12165424

**Published:** 2023-08-21

**Authors:** Richard Conway, Declan Byrne, Deirdre O’Riordan, Bernard Silke

**Affiliations:** Department of Internal Medicine, St James’s Hospital, Dublin 8, D08 NHY1 Dublin, Ireland

**Keywords:** RDW, 30-day mortality, long-term survival, life-year loss

## Abstract

The red cell distribution width (RDW) is the coefficient of variation of the mean corpuscular volume (MCV). We sought to evaluate RDW as a predictor of outcomes following acute medical admission. We studied 10 years of acute medical admissions (2002–2011) with subsequent follow-up to 2021. RDW was converted to a categorical variable, Q1 < 12.9 fl, Q2–Q4 ≥ 12.9 and <15.7 fL and Q5 ≥ 15.7 fL. The predictive value of RDW for 30-day in-hospital and long-term mortality was evaluated with logistic and Cox regression modelling. Adjusted odds ratios (aORs) were calculated and loss of life years estimated. There were 62,184 admissions in 35,140 patients. The 30-day in-hospital mortality (n = 3646) occurred in 5.9% of admissions. An additional 15,086 (42.9%) deaths occurred by December 2021. Admission RDW independently predicted 30-day in-hospital mortality aOR 1.93 (95%CI 1.79, 2.07). Admission RDW independently predicted long-term mortality aOR 1.04 (95%CI 1.02, 1.05). Median survival post-admission was 189 months. For those with admission RDW in Q5, observed survival half-life was 133 months—this represents a shortfall of 5.7 life years (33.9%). In conclusion, admission RDW independently predicts 30-day in-hospital and long-term mortality.

## 1. Introduction

Acute medicine entails the immediate and specialist care of patients presenting with a wide range of medical conditions. By its nature, acute medicine in different institutions may vary in the composition of its patient cohort. This may reflect local demographic and geographical factors and the influence of these on the distribution and frequency of specific diseases within the population. There may also be local protocols, either formal or informal, as to what conditions are deemed appropriate to the remit of acute medicine, and those which are deemed to fall within the purview of other specific specialist services. However overall, key characteristics of acute medicine include a wide breadth of diagnoses and the need for prompt input to improve care. Acute medicine is best viewed as a key specialty in its own right with a skill set that is malleable to local factors.

In 2003, as part of an initiative in acute medicine, our institution opened an Acute Medical Assessment Unit (AMAU). The aim was to improve outcomes for patients requiring urgent or emergency care [1]. The results of this initiative have been reported previously [2,3,4]. Similar to other such initiatives elsewhere, improvements in mortality and increased efficiency have been demonstrated. A predictive algorithm based on admission laboratory tests to predict 30-day in-hospital mortality was developed and validated [5]; one of its significant components was the red cell distribution width (RDW). This algorithm was refined and denominated as the Acute Illness Severity Score (AISS). RDW is a red blood cell (RBC) statistical measure, derived from auto-analysers that describes the variation in red cell width. It is an emerging biomarker of morbidity and mortality in cardiovascular disease [6]. RDW has also been reported to predict outcomes in cancer, infectious diseases and others [7,8,9,10].

The precise explanation as to why discrete populations acquire different RDW with different survival outcomes is unclear. Increasing RDW is a measure of anisocytosis—A greater variation in the red blood cell width; this may result from increased or ineffective production of RBCs or excessive fragmentation or destruction of RBCs [6]. It may represent disparate clinical entities including iron deficiency or megaloblastic anaemia, haemolysis, myelodysplastic syndrome or liver dysfunction.

Our acute medicine initiative dates from the planning stage of 2002, with implementation in 2003 and a 20-year follow-up that includes analysis of the Irish National Death Register to 2021. In this study, we analyse RDW as an outcome predictor for all acute medical admissions to St. James’s Hospital, Dublin, over a 10-year period (2002–2011) and subsequent long-term survival to 2021. 

## 2. Materials and Methods

### 2.1. Background

St James’s Hospital, Dublin, admits acute medical patients as a secondary care level hospital. The institution’s catchment area serves a population comprising 270,000 adults. Acute medical admissions are admitted to an AMAU through the ED. We have reported the systems and outcomes of the AMAU previously [2,3,11]. 

### 2.2. Data Collection

This study utilised anonymous patient databases. The data analysis combined these existing databases including data from each clinical admission. This comprised the resources of the institutions’ electronic patient records, the laboratory data systems, the patient administration system (PAS) and the national Hospital In-Patient Enquiry (HIPE) scheme. HIPE is a database recording coded discharge summaries at a national level from all Irish institutions admitting acute patients [12]. HIPE coding follows The International Classification of Diseases, Clinical Modification. ICD-9-CM was used for both diagnosis and procedure coding from 1990 to 2005 and ICD-10-CM subsequently. In the final data assembly and analysis, variables include the patient’s unique identifier, the consultant in charge, date of birth, gender, area of residence, dates of admission and discharge, principal and up to nine additional secondary diagnoses and principal and up to nine additional secondary procedures. Physiological, haematological and biochemical parameters are accessed from the laboratory systems and included in the analysis. The current study includes all acute medical admissions admitted in the 10 years between 2002 and 2011, with follow-up on the National Death Register to December 2021. The study received institutional ethics approval.

### 2.3. Risk Predictors

Clinical outcomes may be predicted by admission haematological and biochemical values. We have previously derived and applied an AISS [5]. We have demonstrated that the AISS predicts 30-day in-hospital mortality from admission parameters [13]. The AISS calculates a weighted age adjusted score. The AISS stratified outcomes into six risk groups (I–VI) with initial cut-points for 30-day in hospital mortality set at 1, 2, 4, 8 and 16%. The Comorbidity Score was utilised to adjust for comorbidity [14]. The Comorbidity Score was devised by searching ICD codes corresponding to chronic physical and mental health conditions that may limit individuals in performing activities that they would otherwise be expected to be able to perform. These ICD codes were subsequently divided into ten groups reflecting the systems involved: (i) cardiovascular, (ii) respiratory, (iii) neurological, (iv) gastrointestinal, (v) diabetes, (vi) renal, (vii) neoplastic disease, (viii) others (including rheumatological disorders), (ix) ventilatory assistance required and (x) transfusion requirement. To supplement the ICD code diagnostic capture, we further searched other existing databases in our institution to reduce the number of potentially missed cases due to inadequate coding. We searched for evidence of diabetes mellitus using the Diamond database [15], respiratory insufficiency as defined by FEV1 < 2 L using the pulmonary function testing database, elevated troponin (defined as high-sensitivity troponin ≥ 25 ng/L) [16], hypoalbuminaemia (defined as serum albumin < 35 G/dL), anaemia (defined as haemoglobin level < 10 G/dL) and chronic kidney disease (defined as Modification of Diet in Renal Diseases (MDRD) equation < 60 mL/min/1.73 m^2^ [17]. Each component of the score was then weighted according to 30-day in-hospital mortality.

Blood culture categories were defined as (1) no blood culture request (2) negative blood culture and (3) positive blood culture and used as an adjustor in the multiple variable logistic regression model [18].

RDW was divided into quintiles—the split was Q1 < 12.9 fL Q2–Q4 ≥ 12.9 and <15.7 fL and Q5 ≥ 15.7 fL. This resulted in cohort of ranges Q1 12.6 (IQR 12.3, 12.8), Q2–Q4 13.9 (IQR 13.4, 14.6) and Q5 17.2 (IQR 16.3, 18.6). 

### 2.4. Match to Irish National Death Register

Data analysis uses anonymised data, using coded instructions to assemble data for analysis from many discrete files, patients being identified from a unique hospital episode no. or identifier (MRN) and the series number (case-id—related to the admission date/time). Each patient address was accessed once to derive the National Deprivation Index [19,20]. The match to the National Death Registry and PAS record (name, address) involved cleaning of string data and merging of the databases (matched on date of birth and surname). Gender mismatches were deleted, and string-based fuzzy matching techniques calculated a score of text (both name/surname and address) similarity. The match was confirmed by visual inspection. A small number of residual observations (~350) were manually accepted or rejected using online death notices. The information then, on the database, consists of 0/1 and date of death to preserve the anonymous database nature.

### 2.5. Statistical Methods

We calculated descriptive statistics for background demographic data. These were reported using means/standard deviations (SDs), medians/inter-quartile ranges (IQRs) or percentages as appropriate. Chi-square tests were used for comparisons between categorical variables and mortality. 

Logistic regression analysis was used to identify potential mortality predictors. Those that proved to be significant univariate predictors (*p* < 0.1 by Wald test) were tested to ensure that the model included all variables with predictive power. The margins command in Stata was utilised to estimate and interpret adjusted predictions for sub-groups, while controlling for other variables such as time, using computations of average marginal effects. Margins are statistics calculated from predictions of a previously fitted model at fixed values of some covariates and averaging or otherwise over the remaining covariates. In the multivariable logistic regression model, we adjusted for other known predictor variables including AISS [5,18]; Comorbidity Score [14]; Major Disease Categories (MDCs) of neurology (MDC1), respiratory (MDC4), cardiovascular (MDC5); and blood culture category [15]. We employed a logistic model with robust estimation to allow for clustering; the correlation matrix thereby reflected the average discrete risk attributable to each of these predictor variables [5]. Survival regression calculations were undertaken employing Cox’s proportional hazards model for two groups with the assumption of a constant between-group hazard function over time. Testing for the equality of survivor functions was performed with the Log-rank test. Breslow’s method for handling ties with results as hazard ratios—that is, exponentiated coefficients—was utilised. 

We calculated adjusted odds ratios (ORs) and 95% confidence intervals (CIs) for those predictors that significantly entered the model (*p* < 0.10). Statistical significance at *p* < 0.05 was assumed throughout. All data analyses were conducted using the Stata v.17.0 (Stata Corporation, College Station, TX, USA) statistical package. 

## 3. Results

### 3.1. Patient Demographics

There were 62,184 admissions in 35,140 patients between 2002–2011. This included patients admitted directly into the Intensive Care Unit or High Dependency Unit. The proportion of males was 48.2%. The median (IQR) LOS was 5.0 (2.0, 9.8) days. The median (IQR) age was 61.9 (41.7, 61.9) years, with the upper 10% boundary at 84.2 years.

### 3.2. Demographics Related to 30-Day In-Hospital Mortality

The 30-day in-hospital mortality (n = 3646) by patient was 10.4% or by admission 5.9%. Patients in RDW Q1 were younger, 49.5 (31.2, 69.4) years, with the shortest LOS 4.3 (1.8, 8.9) (Table 1). Patients in the other quintiles were older. Persons in Q5 were aged 67.5 (48.4, 79.5) years and had the longest LOS 8.2 (3.9, 17.5) days. RDW Q1 cohort, in addition to being younger, had lower grades of AISS, Charlson Index and Comorbidity Score. 

### 3.3. RDW as a Predictor of 30-Day In-Hospital Mortality

In univariate analyses, RDW significantly predicted 30-day in-hospital mortality, OR 2.62 (95%CI 2.46, 2.80). RDW predicted 30-day in-hospital mortality with AUROC 0.87; see Figure 1.

In the multivariable model, RDW significantly predicted 30-day in-hospital mortality, OR 1.93 (95%CI 1.79, 2.07). The model adjusted 30-day in-hospital mortality by RDW quintile was Q1 2.5% (95%CI 2.3, 2.8), Q2–Q4 4.4% (95%CI 4.3, 4.6) and Q5 7.5% (95%CI 7.1, 7.9). RDW predicted 30-day in-hospital mortality increased for each quintile with increasing Comorbidity Score (Figure 2). The overall model predicted 30-day in-hospital mortality with AUROC 0.84 (95%CI 0.83, 0.85).

### 3.4. RDW as a Predictor of Long-Term Mortality

The long-term follow-up revealed a total of 15,086 (42.9%) additional deaths to have occurred. The average duration of follow-up was 122 months (IQR 86, 156). 

In univariate analyses RDW was a significant predictor of long-term mortality OR 1.13 (95%CI 1.11, 1.14); see Figure 3. RDW predicted long-term mortality with AUROC of 0.69; see Figure 2. 

In the multivariable model, RDW significantly predicted long-term mortality, OR 1.04 (95%CI 1.02, 1.05); see Table 2. Other predictors of long-term mortality were Comorbidity Score OR 1.11 (95%CI 1.10, 1.12), Charlson Index OR 1.20 (95%CI 1.15, 1.24) and a cardiovascular MDC OR 1.17 (95%CI 1.09, 1.26); see Table 2. There was a significant interaction between older age and frequent readmission in the multivariable model, OR 1.21 (95%CI 1.18, 1.24) (*p* < 0.001). The overall model predicted long-term mortality with an AUROC of 0.78 (95%CI 0.76, 0.78).

### 3.5. Survival Analysis of Long-Term Mortality

Those who died during long-term follow-up were of median age 75.4 years (IQR 63.7, 82.8) at time of admission, and median age at death was 80 years (IQR 69, 87). The overall survival half-life, after hospital discharge for the entire RDW cohort, was 189 months (15.8 years). The survival for RDW Q5 at time of hospital admission was significant worse at 133 months (11.1 years); see Figure 4. The latter population, at time of hospital admission, had a median age of 67.5 years (48.4, 79.5), and the CSO Irish Life Tables (2010–2015) projected survival at this age of 16.8 years. The observed post-hospital discharge survival thus represented a shortfall of 5.7 life years (33.9%). 

## 4. Discussion

This study demonstrates that RDW at the time of acute medical admission predicts both 30-day in-hospital and subsequent long-term community mortality. RDW was a stronger predictor of 30-day in-hospital mortality (OR 2.62 (95%CI 2.46, 2.80)) compared with longer-term community mortality (OR 1.13 (95%CI 1.11, 1.14)). The complete multivariable model had a superior prediction for 30-day in-hospital mortality but continued to perform well to predict long-term mortality (AUROC 0.84 (95%CI 0.83, 0.85) vs. AUROC 0.78 (95%CI 0.76, 0.78)). Those in RDW Q5 had a survival half-life of 11.1 years post-discharge compared to overall post-discharge survival of 15.8 years. This represented a shortfall of 5.7 life years (33.9%) in projected life expectancy for individuals of median age 67.5 years at admission compared to age and gender matched peers. 

RDW is a quantitative measure of variability in size of circulating erythrocytes—effectively the coefficient of variation of the erythrocyte MCV. The rationale for why RDW would predict outcomes is not intuitively obvious, although systemic factors that alter erythrocyte homeostasis such as inflammation and oxidative stress have been implicated [7]. A study using National Health and Nutrition Examination Survey (NHANES) data of 8175 adults aged 45 years or older reported that for every 1% increment in RDW there was an increase in all-cause mortality of 22% [21]. RDW remained strongly associated with mortality even in non-anaemic subjects; those without iron, folate or vitamin B12 deficiency; and those in the normal RDW range (11–15%) [21]. This association with increased mortality persisted after adjustment for major age-associated diseases, education, BMI, smoking status, hospitalisations and renal function. 

Our data confirm these figures in a 10-year cohort and provide 10-year community follow-up; it is clear from the survival curves that mortality increased, from low to high quintiles of RDW, even after adjustment for admission AISS [5,22], Comorbidity Score [14,20] and blood culture category [23]. While its original role was as a marker in anaemia, there are several mechanisms by which RDW may be associated with mortality. The association of RDW with mortality may reflect its role as a marker of general health status. Low-grade inflammation and erythropoietic variability may lead to greater heterogeneity in erythrocyte size (younger erythrocytes are usually larger and more variable in size than older), and oxidative stress might also contribute to anisocytosis. While erythrocytes have tremendous antioxidant capacity and serve as the primary “oxidative sink”, they are prone to oxidative damage that may reduce cell survival in a variety of pathological states [21]. Additionally, elevated RDW might indicate an impaired or inefficient production of red blood cells, potentially pointing to a disparate collection of underlying health conditions that could contribute to mortality. Furthermore, RDW variability at a population level may reflect a difficult to quantify reflection of general health status. Further research is required to fully elucidate the precise mechanisms through which RDW may be associated with mortality.

The question, however, arises as to whether RDW can be regarded as a genuine risk factor or merely an epiphenomenon of underlying biological processes. There is increased heterogeneity of RBC as we age due the decreased erythrocyte deformability. Lippi et al. showed a strong dependence between RDW and age in 1907 healthy people, with the proportion with RDW > 14.6% (as a population cut-point risk marker) increasing from 6% aged <41 years to 75% >90 years [24]. This effect of increasing age on RDW appears more pronounced in women [9]. However, RDW has been shown to be of prognostic significance in many different disease processes, including myocardial infarction, congestive heart failure, stroke, pulmonary embolism, cancer, inflammatory bowel disease, major trauma and sepsis [7,8,9]. Our data are derived from a local hospital catchment area population with high intrinsic rates of deprivation [20,25,26]. Assessments of survival must take account of the considerably younger age of this population at the time of hospital admission. Individuals from areas of high deprivation were younger on admission, median age 61.3 years (IQR 41.2, 76.4), compared to low-deprivation areas at 75.2 years (IQR 60.0, 82.5). This needs to be considered when projecting long-term survival; nevertheless, the pattern of other studies of decreased survival as RDW increased was confirmed.

There are data in the existing literature on the impaired general long-term survival following an unplanned hospital admission. Quinn et al. reported on the risk of death within 5 years of first hospital admission for older adults [27]. After the first unplanned hospital admission, 39.7% had died by 5 years in this Canadian study [27]. Flojstrup et al. using data from the Danish National Registry at population age range of 75–79 years, reported mortality rates of for males of 29.3% and females of 25.8% at 1 year following an acute hospitalisation [28]. Clark and Moore et al. audited Scottish hospitals on two nominated dates in different years and reported 1-year mortality in the age range 75–79 years to be 33% but with further increases at higher ages [29,30]. 

The emergency medical admission has been defined as a sentinel event that is “any unanticipated event in a healthcare setting that results in death or serious physical or psychological injury to a patient, not related to the natural course of the patient’s illness”. Of course, population level data may not reflect an individual’s risk, but the aggregate population effect is such that the 24 h acute-take size can be estimated with reasonable certainty. Once an admission occurs, it entails a significant and measurable risk, but the translation of earlier onset of the emergency admission to shortened life expectancy is not obvious. Our data shows that RDW is an additional parameter that can be employed to measure risk. The upper RDW quintile appeared to shorten life expectancy by 5.7 life years (33.9%), compared with the Irish population Irish Life Tables (2010–2015) projected survival at age 68 years of 16.8 years.

Previous research has delineated the importance of other predictors of outcomes following acute medical admission. Short-term mortality, generally defined as 30-day in-hospital mortality, has been studied to a far greater extent than long-term mortality. The reason for this discrepancy is most frequently a pragmatic one—linking long-term mortality data to an acute hospital dataset involves complex data collection and analytics. In our own population, the most consistent predictors of 30-day in-hospital morality are the severity of the index acute illness and the comorbidity burden at the time of index hospital admission [11,14]. Other work has assessed the additional predictive capability of individual biomarkers and demonstrated predictive value for troponin, B-type natriuretic peptide (BNP), blood culture results and serum potassium and sodium among other parameters [23,31,32,33]. The value of these markers outside of their traditionally considered disease specific prognostic value has been demonstrated, with effects seen across acute medical diagnoses. This study illustrates that RDW can be considered as another potential signal of future short-term increased risk of negative outcomes during an acute medical admission. Much less research has explored the potential prediction of long-term mortality following acute medical admission and the consideration of how factors identified at the time of a hospital admission may influence or predict subsequent outcomes potentially many years later. We have previously shown that logical parameters of age and comorbidity at the time of the index admission are predictors of long-term mortality [34]. Rather more enigmatically, we identified the severity of the index hospital admission as a predictor of long-term outcomes [34]. The reasons for this association remain to be fully elucidated but may reflect either a deficit of inherent physiologic adaptive mechanisms or an impact on the ability to bounce back to baseline following more severe illness events.

Our study adds significantly to the literature in this area. We have used a comprehensive and rigorous dataset to evaluate the predictive capability of RDW in acute medical admissions. In terms of short-term mortality (30-day in-hospital mortality), we add to the existing literature as the detail and completeness of our dataset allows us to control for multiple other potential confounders and effect modulators. We further extend the literature through an evaluation of overall mortality prediction across all acute medical admission diagnoses; many previous studies were confined to single diseases or disease areas. The robustness of the predictive value of RDW across the medical admission group as a whole argues against there being one explanation such as inflammation or anaemia for the association with mortality—and indeed parameters reflecting these were not significant predictors in our model. Our addition of long-term morality data, tied to a comprehensive admission dataset is unique in the literature and confirms the long-term predictive value of admission RDW values. 

As with any study, there are potential limitations to the current work. This was a single-centre study—our findings require validation in other settings and countries to confirm external validity. We employed a comprehensive dataset and controlled for multiple other known outcome predictors; however, it is possible that unmeasured confounders remain which could explain our findings. It is important to reflect on the fact that we report an association between a laboratory value and outcome. This does not imply that this is a parameter that is amenable to intervention, or indeed that it should be targeted as such. We have previously demonstrated the potential folly, and indeed paradoxical negative effects, of targeting such “numbers” [35].

In conclusion, we have demonstrated that admission RDW predicts both 30-day in-hospital and long-term community mortality. For those in the highest quintile of RDW, a dramatic 5.7-year shortfall in subsequent life expectancy exists. The clinical utility of this information at present is limited to assisting in a framework around prognostic discussions. There is the possibility that outcomes may be amenable to intervention, but this requires further study to elucidate.

## Figures and Tables

**Figure 1 jcm-12-05424-f001:**
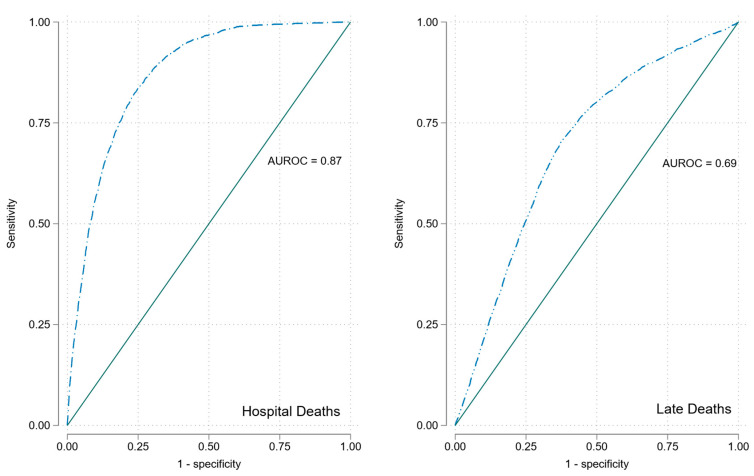
ROCs for RDW for 30-day in-hospital and long-term mortality.

**Figure 2 jcm-12-05424-f002:**
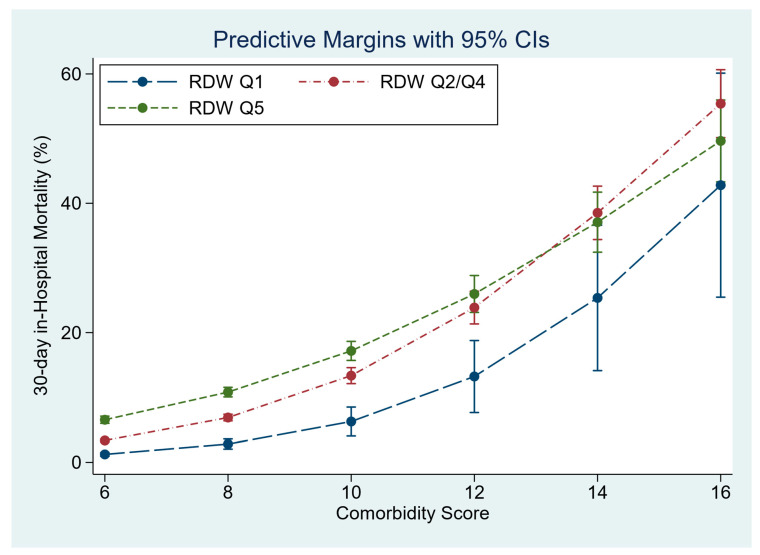
The influence of admission RDW, at different Comorbidity Scores, on 30-day in-hospital mortality.

**Figure 3 jcm-12-05424-f003:**
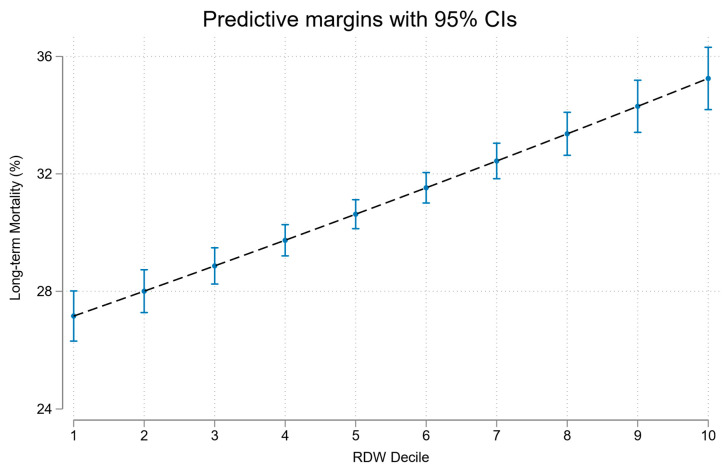
Association of RDW decile with long-term mortality.

**Figure 4 jcm-12-05424-f004:**
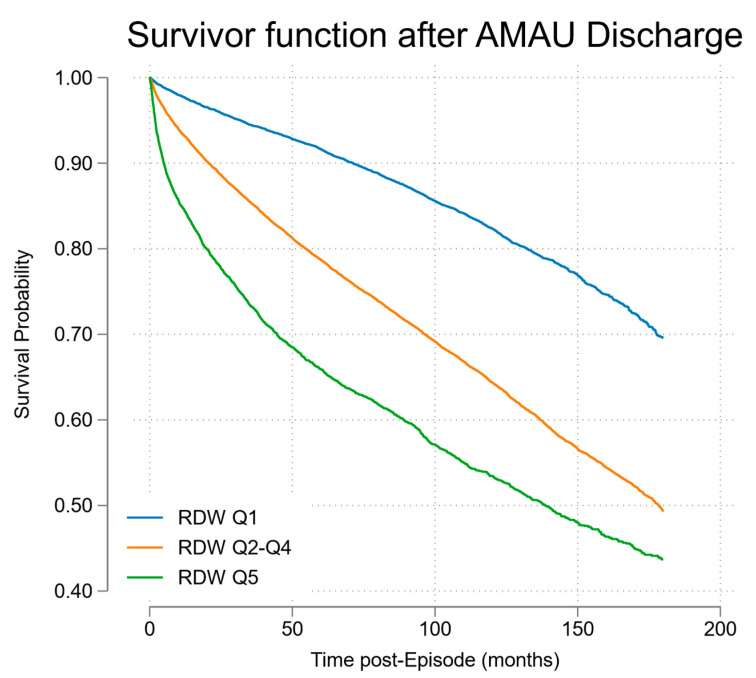
Cox proportional hazards model of survival post-discharge stratified by RDW quintile.

**Table 1 jcm-12-05424-t001:** 2002–2011 Admission Characteristics.

Variable		RDW Q1	RDW Q2–Q4	RDW Q5	*p*-Value
N		12,240	38,673	11,271	
Age (years)		49.5 (31.2, 69.4)	66.6 (46.4, 79.1)	67.5 (48.4, 79.5)	<0.001
LOS (days)		4.3 (1.8, 8.9)	5.9 (2.3, 12.9)	8.2 (3.9, 17.5)	<0.001
Gender	Male	6233 (50.9%)	18,368 (48.5%)	5392 (47.8%)	<0.001
	Female	6007 (49.1%)	19,469 (51.5%)	5879 (52.2%)	
Acute Illness	1–3	6933 (56.6%)	8948 (23.1%)	350 (3.1%)	<0.001
Severity Groups	4	2486 (20.3%)	6308 (16.3%)	1146 (10.2%)	
	5	1777 (14.5%)	8524 (22.0%)	1940 (17.2%)	
	6	1044 (8.5%)	14,893 (38.5%)	7835 (69.5%)	
Charlson Index	0	6969 (57.0%)	15,581 (41.3%)	3357 (29.9%)	<0.001
	1	2954 (24.2%)	10,023 (26.6%)	2808 (25.0%)	
	2	2293 (18.8%)	12,081 (32.1%)	5062 (45.1%)	
Comorbidity	<6	8609 (70.3%)	19,905 (51.5%)	4396 (39.0%)	<0.001
Score	6 < 10	3146 (25.7%)	15,100 (39.0%)	5150 (45.7%)	
	≥10	485 (4.0%)	3668 (9.5%)	1725 (15.3%)	
Blood Culture	0	9243 (75.5%)	27,738 (73.3%)	7635 (67.7%)	<0.001
Groups	1	2584 (21.1%)	8312 (22.0%)	2822 (25.0%)	
	2	413 (3.4%)	1787 (4.7%)	814 (7.2%)	

**Table 2 jcm-12-05424-t002:** Multivariable logistic regression model of long-term mortality.

Variable	OR	Std. Err.	z	*p* > |z|	[95% Conf. Interval]	
RDW	1.04	0.01	5.3	0.00	1.02	1.05
Comorbidity Score	1.11	0.01	21.4	0.00	1.10	1.12
Charlson Index	1.20	0.02	9.9	0.00	1.15	1.24
Readmission No.	1.03	0.01	4.5	0.00	1.02	1.05
Respiratory	0.92	0.03	−2.4	0.02	0.86	0.98
Cardiovascular	1.17	0.04	4.2	0.00	1.09	1.26
Neurology	0.90	0.03	−2.8	0.01	0.83	0.97
Readmission/Older	1.21	0.01	16.4	0.00	1.18	1.24

## Data Availability

No further data are available for sharing. All pertinent data have been published in this manuscript.

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
