# Peer review of "Red Cell Distribution Width as a Prognostic Indicator in Acute Medical Admissions"

_jcm, 2023, doi:10.3390/jcm12165424_

Round 1
Reviewer 1 Report
Red cell distribution width (RDW) is a parameter that reflects the variability of the size of red blood cells (anisocytosis). Elevated RDW is a result of erythrocytes production dysfunction related to a deficiency of folic acid, iron, vitamin B12 or ongoing inflammation as well as increased destruction of erythrocytes e.g. in the course of hemolysis. Previous studies have indicated the predictive ability of RDW in various cardiovascular disorders or acute renal injury (1).
This is an interesting article, but it is worth supplementing the discussion with potential reasons for the increased RDW values ​​(1).
Reference:
1. Doi: 10.5603/CJ.a2019.0020
Thank you.
Author Response
Thank you, we have added the following “While its original role was as a marker in anaemia, there are several mechanisms by which RDW may be associated with mortality The association of RDW with mortality may reflect its role as a marker of general health status. Low grade inflammation and erythropoietic variability may lead to greater heterogeneity in erythrocyte size (younger erythrocytes are usually larger and more variable in size than older) and oxidative stress might also contribute to anisocytosis. While erythrocytes have tremendous antioxidant capacity and serve as the primary “oxidative sink”, they are prone to oxidative damage that may reduce cell survival in a variety of pathological states (18). Additionally, ele-vated RDW might indicate an impaired or inefficient production of red blood cells, po-tentially pointing to a disparate collection of underlying health conditions that could contribute to mortality. Furthermore, RDW variability at a population level may reflect a difficult to quantify reflection of general health status. Further research is required to fully elucidate the precise mechanisms through which RDW may be associated with mortality.”
Reviewer 2 Report
Overall this manuscript is well-written and had clear and comprehensive descriptions. I suggest the authors to address more on the "novelty" of this study, since the correlation between RDW and outcomes of the patientswith acute illness was well-validated and many literature had similar results.
Author Response
Thank you, we have added the following “Our study adds significantly to the literature in this area. We have used a compre-hensive and rigorous dataset to evaluate the predictive capability of RDW in acute medical admissions. In terms of short-term mortality (30-day in-hospital mortality), we add to the existing literature as the detail and completeness of our dataset allows us to control for multiple other potential confounders and effect modulators. We further ex-tend the literature by an evaluation of overall mortality prediction across all acute medical admission diagnoses, many previous studies were confined to single diseases or disease areas. The robustness of the predictive value of RDW across the medical admis-sion group as a whole argues against there being one explanation such as inflammation or anaemia for the association with mortality – and indeed parameters reflecting these were not significant predictors in our model. Our addition of long-term morality data, tied to a comprehensive admission dataset is unique in the literature and confirms the long-term predictive value of admission RDW values.”
Reviewer 3 Report
In this study the authors aimed to evaluate RDW as a predictor of outcome following acute medical admission. The manuscript is well written and the main idea and methods are sound. However there are some room for improvement. First of all conclusion must be more evolved (more then one sentence) with more emphatic highlights of this study and their practical use. It would be usefull to know whether RDW was different among different pathologies (different admission criteria). Moreover, since RDW was shown to be elevated in inflamation it would be useful to include parameters of inflmaation in the analysis (if they are not available please state in limitations). Also, some of the parameters regarding to anemia (Hgb levels, MCV, ferritin, paramters of hemolysis) could be taken into account.
Author Response
Thank you, we have added the following to the conclusion “In conclusion, we have demonstrated that admission RDW predicts both 30-day in-hospital and long-term community mortality. For those in the highest quintile of RDW, a dramatic 5.7 year shortfall in subsequent life expectancy exists. The clinical utility of this information at present is limited to assisting in a framework around prognostic discussions. There is the possibility that outcomes may be amenable to in-tervention but this requires further study to elucidate.”
Regarding including parameters of inflammation and anaemia in the analysis. These were assessed for inclusion in the multivariable modelling but were not significant predictors. We have added to the discussion “The robustness of the predictive value of RDW across the medical admission group as a whole argues against there being one explanation such as inflammation or anaemia for the association with mortality – and indeed parameters reflecting these were not significant predictors in our model.”
Round 2
Reviewer 3 Report
I am fine with this version of manuscript.
Author Response
Thank you.
As requested by editor we have added ROC curves for RDW